# Momentary Physical Activity Co-Occurs with Healthy and Unhealthy Dietary Intake in African American College Freshmen

**DOI:** 10.3390/nu12051360

**Published:** 2020-05-09

**Authors:** Jaclyn P. Maher, Meghan Harduk, Derek J. Hevel, William M. Adams, Jared T. McGuirt

**Affiliations:** 1Department of Kinesiology, University of North Carolina Greensboro, Greensboro, NC 27412, USA; mlharduk@uncg.edu (M.H.); djhevel@uncg.edu (D.J.H.); wmadams@uncg.edu (W.M.A.); 2Department of Nutrition, University of North Carolina Greensboro, Greensboro, NC 27412, USA; jtmcguir@uncg.edu

**Keywords:** ecological momentary assessment, exercise, eating, fruit and vegetables, sugar-sweetened beverages

## Abstract

**Background:** Research investigating interrelations between physical activity and dietary intake has primarily used retrospective, summary-based measures of behavior subject to increased recall bias. This study used ecological momentary assessment (EMA) methods with accelerometry to determine within-day, momentary associations between physical activity and dietary intake behaviors in African American college freshmen. Methods: Participants (*N* = 50) completed a dietary EMA protocol that assessed food/fluids consumed over the past 2 h at five random times per day and wore an activPAL accelerometer for 7 days to measure physical activity. Physical activity was operationalized as step counts in the 2 h prior to the EMA prompt (matching the EMA recall window). **Results:** On occasions when participants took more steps than was typical for them in the 2 h prior to the EMA prompt, they were more likely to consume sugar-sweetened beverages (OR = 1.37, *p* < 0.001), water (OR = 1.28, *p* < 0.001), fruit (OR = 1.44, *p* < 0.001), vegetables (OR = 1.19, *p* = 0.02), and fried fast food (OR = 1.21, *p* = 0.04) over that same time. **Conclusion:** Momentary physical activity co-occurred with momentary consumption of both healthy and unhealthy dietary intake. These behavioral interrelations suggest potential implications for obesity risk and multiple health behavior change interventions in young adult African Americans.

## 1. Introduction

Obesity and obesity-related health disparities exist between African American adults and their White counterparts. Approximately 48.0% of non-Hispanic Black adults are affected by obesity, whereas only 34.6% of non-Hispanic White adults are affected by obesity, placing African Americans at greater risk for a host of other health issues including heart disease, stroke, and diabetes [1].

A particularly important transition period with implications for weight-related health behavior engagement throughout adulthood is the first semester of college. During this period, individuals are likely to experience significant changes in their roles and responsibilities, resulting in independence, shifting priorities, motivation, and goal pursuits [2]. Physical activity and dietary intake are weight-related behaviors known to change across the transition into college, with declines in physical activity and increases in unhealthy dietary behaviors (e.g., skipping breakfast, fast food consumption) [3,4,5,6]. However, it is unclear the extent to which occasions of physical activity co-occur with dietary intake behaviors during this period.

A recent review reported equivocal findings regarding the extent to which physical inactivity and lack of fruits and vegetables co-occur [7]. For instance, one study of Scottish participants (*n* = 6574, 16–64 years) found that consuming less than five daily servings of fruits and vegetables and engaging in less than 30 min of physical activity, five times per week co-occurred in only 9.7% of adults [8]. Whereas in a sample of British participants (*n* = 11,492, 16–64 years), 47.3% of men and 53.75% of women indicated the co-occurrence of inadequate fruit and vegetable intake and insufficient physical activity levels [9]. A cluster analysis of risk behaviors in university students (*n* = 410) found that students displaying high psychological stress, low physical activity levels, low fruit and vegetable intake, and who were occasional/regular smokers clustered together in one group, whereas students with low psychological stress, high physical activity levels, and high fruit and vegetable intake clustered together in another group [10].

The uncertainty surrounding the associations between physical activity and dietary intake could be due to the limitations of previous research. First, previous research has largely captured physical activity and dietary behaviors using self-reported, summary-based measures designed to capture individuals’ typical or usual behaviors on macro-timescales (e.g., weeks, months, years) [7]. While this information does provide some useful insights regarding associations between physical activity and diet, this approach is limited in that it (1) is prone to recall errors and biases which may result in inaccurate reporting of these behaviors [11] and (2) ignores within-person fluctuations in physical activity and dietary intake behaviors [12]. Capturing the dynamics of these health behaviors on micro-timescales (e.g., minutes, hours, days) may provide useful information about the extent to which these behaviors co-occur in the context of everyday life. Ecological momentary assessment (EMA) methods can capture these dynamics. EMA is a real-time data capture methodology designed to repeatedly assess a phenomenon of interest (e.g., dietary intake) as it unfolds over time in real-world environments [13,14]. Therefore, data collected through EMA can reduce the potential for recall biases and errors associated with traditional retrospective reports because the assessment occurs close in time to when the phenomenon of interest occurs [14,15]. Furthermore, the repeated assessments of EMA are well suited to capture the within-person dynamics of a phenomenon of interest and related constructs as they ebb and flow over time and contexts [14]. Additionally, EMA has the potential to provide more ecologically valid assessments of a phenomenon of interest as it reflects experiences and behaviors encountered in everyday life [14]. Finally, EMA prompts delivered through a mobile or smart device are date and time stamped, which allows for data to be paired with other ambulatory assessment methods such as accelerometers.

EMA methods have previously been used to explore relations between daily physical activity and dietary intake, but findings regarding associations between these health behaviors are mixed. Dunton et al. found that daily moderate- to vigorous-intensity physical activity was positively associated with daily glycemic load and percent of energy from carbohydrates, but negatively associated with daily percent of energy from fats among a sample of children (52% Hispanic) [16]. A study of overweight and obese adults (72% White) in a weight loss intervention found that momentary total physical activity was negatively associated with dietary lapses reported through EMA [17]. Despite evidence that EMA is a valid tool for assessing momentary healthy (e.g., fruits and vegetables) and unhealthy (e.g., salty snacks) eating behaviors among a diverse sample of college students [18], EMA has yet to be used in conjunction with accelerometers to examine within-person, momentary associations between physical activity and dietary behaviors in college students. Understanding these associations in a population of African American college students, who are at disproportionate risk for obesity and other chronic conditions later in life [1], has important implications for interventions designed to prevent weight gain or promote weight loss in this population.

Also, previous research on co-occurrence between dietary intake and physical activity behaviors has mostly focused on fruit and vegetable intake (e.g., [7]), which could be too narrow of a focus given the health risks associated with other foods like added sugars and fried foods. It is unclear the extent to which physical activity and a variety of healthy and unhealthy dietary behaviors co-occur, and more research is needed to understand this co-occurrence using EMA methods.

Thus, this study was designed to determine within-person, momentary associations between physical activity and dietary intake behaviors. A 7 day study integrating EMA to assess healthy and unhealthy dietary intake and accelerometry to measure physical activity was conducted among African American college freshman. We hypothesized that momentary physical activity would be positively associated with both momentary healthy and unhealthy dietary intake behaviors among African American college freshman.

## 2. Materials and Methods

### 2.1. Participants

Recruitment advertisements about the Freshman Reports of Eating, Exercise, Sitting, and Health (FRESH) study were distributed via email to all first-semester freshmen that had previously identified as African American/Black on their college admissions paperwork at a public university in the southeastern United States. Inclusion criteria for the study included (a) identifying as African American/Black, (b) a first-semester college freshman, and (c) 18–25 years of age. Participants were excluded if they (a) had a current injury that prevented their ability to be physically active, (b) made changes to their diet or activity patterns within the week prior to testing, (c) were actively trying to lose or gain weight within the week prior to testing, (d) had gastrointestinal or digestive tract surgery or (e) were pregnant.

### 2.2. Procedures

In total, this study lasted 7 days. On Day 1, individuals meeting eligibility criteria attended an introductory session where they were familiarized with the study procedures. Once consent was provided, research staff trained participants on how to use a Moto G4 (Motorola Inc., Chicago, IL, USA) smartphone with an Android (Google Inc., Mountain View, CA, USA) operating system and an activPAL3 accelerometer (PAL Technologies Ltd., Glasgow, Scotland, UK).

Participants were trained to complete a signal-contingent ecological momentary assessment (EMA) protocol on the assigned study phone. Smartphones were preloaded with the commercially available mobile application movisensXS (movisens, GmbH, Karlsruhe, Germany). Participants were randomly prompted within 5, 1 h segments (i.e., 9:30–10:30 AM, 12:30–1:30 PM, 3:30–4:30 PM, 6:30–7:30 PM, 9:30–10:30 PM) throughout the day for the study period. As a result, participants could receive a maximum of 35 prompts during the study. Depending on the time of the introductory appointment on Day 1, participants may not have received all 5 EMA prompts on Day 1. Participants were not aware of this specific prompting schedule and were only informed that prompts would occur randomly between 9:30 AM and 10:30 PM.

An auditory signal emitted by the phone would prompt the participants to complete the EMA questionnaire. From the first auditory signal, participants were given 15 min to respond to the EMA questionnaire. If participants did not begin completing the EMA questionnaire within the 15 min window, then the questionnaire became unavailable and was marked as missing. Participants did receive reminder auditory signals within those 15 min, at 5 min intervals to alert participants to the EMA questionnaire in the event that they missed or were unable to answer the original prompt. Each EMA questionnaire consisted of up to 14 items and assessed health behavior engagement in the two hours prior as well as current feelings of stress, hunger, and thirst. Each EMA questionnaire took approximately 2–3 min to complete.

At the introductory session, participants were also trained on how to wear the activPAL3 accelerometer. Participants were instructed to wear the activity monitor on the anterior thigh, approximately 3–4 inches above the kneecap. The activity monitor was sealed in poly-tubing to allow for continuous wear even while being submerged under water. The activity monitor was secured using hypafix tape. Participants were instructed to wear the accelerometer during all waking and sleeping hours. In addition, participants kept an activity log to record any time the monitor was not being worn. Participants began wearing the activPAL3 accelerometer during the introductory session.

Prior to leaving the introductory session, participants completed baseline health history and demographic questionnaires. Anthropometric measures were taken in duplicate by a trained research assistant. Participants began receiving EMA prompts on their assigned study phone after leaving the introductory appointment. Participants were instructed to go about their normal daily routines for the duration of the study while wearing the accelerometer and answering the EMA questionnaires. On Day 8, the participants met with the research team to return their study equipment. All subjects gave their informed consent for inclusion before they participated in the study. The study was conducted in accordance with the Declaration of Helsinki, and the protocol was approved by the Ethics Committee of the University of North Carolina Greensboro (Institutional Review Board number 18-0269).

### 2.3. Measures

#### 2.3.1. Dietary Intake

Food consumption was first determined by asking participants whether they had anything to eat within the past 2 h, with a ‘Yes’ or ‘No’ response. If indicated ‘Yes,’ a follow-up question asked participants to indicate the types of foods consumed within the past 2 h. Response options included ‘fruits,’ ‘vegetables,’ ‘fast food,’ ‘chips or fries,’ ‘pastries or sweets,’, and ‘whole wheat foods.’ If the participants indicated they had consumed fast food, they received a follow-up question asking a yes/no question about whether that fast food was fried. Whole wheat food data are excluded from the results due to concerns over college students’ ability to identify whole wheat foods [19,20]. Fruits, vegetables, fried fast food, chips/fries, and pastries/sweets were prioritized for data analysis because these foods represent commonly consumed foods among adults which are associated with either increased weight gain and obesity risk (i.e., fried fast food, chips/fries, and pastries/sweets) [21,22] or weight maintenance and decreased obesity risk (i.e., fruits, vegetables) [23].

Fluid consumption was assessed using a single item, where participants were asked whether they had anything to drink within the past 2 h with a ‘Yes’ or ‘No’ response. If participants indicated ‘Yes,’ a follow-up question asked participants to indicate what type of fluid was consumed. Response options included ‘water,’ ‘soda or sweet tea,’ ‘diet soda,’ ‘energy drink,’ ‘sports drink,’ ‘coffee,’ ‘milk,’ ‘fruit juice,’ and ‘alcohol.’ Based on EMA responses, a sugar-sweetened beverage variable was created. If participants indicated they had consumed soda or sweet tea, energy drink, sports drink, or fruit juice, sugar-sweetened beverage consumption was indicated at that EMA prompt. Water and sugar-sweetened beverage consumption were prioritized for data analysis because these fluids represent commonly consumed fluids among adults that have been associated with increased weight gain and obesity risk (i.e., sugar-sweetened beverages) [22,24,25] or weight maintenance and decreased obesity risk (i.e., water) [26].

EMA items and response options pertaining to food and fluid consumption were adapted from previous EMA research assessing dietary intake behaviors in children [27] and college students [18]. Due to the need to reduce the potential for participant fatigue from lengthy questionnaires or items that are difficult for participants to answer (e.g., food quantity), EMA signal-contingent assessments of dietary intake typically focus on consumption or frequency of consumption of different groups or types of food as opposed to quantity consumed or preparation method of different foods [28]. Previous research in college students has documented that EMA assessments of food groups or types of foods (e.g., entrees, fruits/vegetables, salty foods, sweets) have had high match rates with consumption reported in 24 h dietary recalls [18].

#### 2.3.2. Physical Activity

The activPAL3 accelerometer was used to assess physical activity. Specifically, physical activity was operationalized as the number of steps taken in the 2 h before the EMA prompt. ActivPAL data were recorded in 15 s epochs and steps were determined using activPAL’s proprietary algorithms. Steps represent a basic unit of locomotion behavior, are easy to measure, and intuitive [29]. Additionally, step counts have been linked with cardiovascular disease events, type 2 diabetes mellitus onset, and all-cause mortality among adult populations [30,31,32]. By conceptualizing physical activity as steps counts, it increases the potential for these findings to inform health promotion programs and policies at universities and colleges [33]. The activPAL3 monitor has previously been validated as a measure of stepping behavior (including step counts) in samples ranging from age 6 to 65 years [34]. Activity monitor logs were used to determine non-wear, which was defined as wearing the activity monitor for less than 60 min in the 2 h prior to the EMA prompt. If participants did not wear the activity monitor for at least 60 min, the occasion was not considered valid and excluded from data analysis.

#### 2.3.3. Temporal Variables

The activPAL3 and EMA data were date and time stamped to determine the time of day and the day of the week. Based on when the participant answered the EMA prompt, the data were coded for day of week (weekday versus weekend day). The time of day was coded as ‘morning’ (9:30–10:30 AM), ‘afternoon’ (12:30–1:30 PM, 3:30–4:30 PM), and ‘evening’ (6:30–7:30 PM, 9:30–10:30 PM).

#### 2.3.4. Demographic Characteristics

Anthropometric measures of height and weight were taken using a Seca stadiometer (Model 216, Chino, CA, USA) and Tanita scale (WB-800S Plus, Tanita Corporation, Tokyo, Japan), respectively. Height (m) and weight (kg) measurements were used to calculate a participant’s BMI (kg/m^2^). Age and race were self-reported as part of the demographic questionnaire during the introductory appointment.

### 2.4. Data Analysis

#### 2.4.1. Data Preparation

All dietary intake variables were coded as “1” for having consumed that food/drink in the past two hours or “0” for not having consumed that food/drink in the past two hours. BMI was grand mean centered. Sex was coded as male “0” and female “1”. Days of the week were coded as weekend “1” or weekday “0”. Time of day was coded as morning “0”, afternoon “1”, and evening “2”.

To examine between- and within-person associations between physical activity and food and fluid consumption, the physical activity variable (i.e., step counts) was disaggregated to create between- and within-person variables also known as partitioning the variance. Therefore, the between-person effect represents each individual’s mean deviation from the group mean (i.e., differentiates individuals who took more or less steps, on average) and the within-person effect represents the occasion-level deviation from one’s own mean (i.e., differentiates occasions where a participant took more or less steps than was typical for them).

#### 2.4.2. Multilevel Modeling

Multilevel logistic regressions were estimated to determine the odds of consuming specific types of food and beverages while accounting for the clustering of observations within individuals. Regarding food consumption, separate models were estimated to determine the odds of consuming fruits, vegetables, fried fast food, chips/fries, and sweets/pastries. Regarding fluid consumption, separate multilevel models were estimated to determine the odds of consuming water and sugar-sweetened beverages. Thus, in separate multilevel logistic models, the odds of consuming specific types of food and beverages in the two hours prior to the EMA prompt were regressed on step counts in the two hours prior to the EMA prompt in order to match the EMA recall window, time of day, day of week, sex, and BMI. To facilitate interpretation of odds ratios in multilevel logistic regressions, step counts were rescaled by 1000. A random intercept was included in all models but the random effect for within-person step counts was not included due to convergence issues in the multilevel logistic regression models.

This study’s multilevel analysis is focused on the momentary, within-person association between physical activity and dietary behaviors. Therefore, the power of the statistical test depends on total number of observations (adjusted for within-person correlations from occasion to occasion). Due to the intensive sampling of this study design (i.e., up to 35 observations per person), this study is likely powered to detect small within-person associations. The research team did not conduct an a priori power analysis; however, the number of observations in this study is approximately equivalent to or greater than several other studies examining within-person fluctuations in dietary intake [28,35].

## 3. Results

### 3.1. Participants

Approximately 70 individuals contacted the research team and expressed an interest in participating. Of those 70 individuals, 64 were determined to be eligible to participate. Individuals who were ineligible were so because they indicated trying to gain or lose weight within the past week (*n* = 6). Of those who were eligible to participate, 61 were scheduled for their introductory appointment. Of those scheduled, nine individuals did not show up to their appointment. Additionally, two withdrew from the study shortly after they began the study, as these participants felt too busy to complete the study procedures.

In total, 50 participants who self-identified as African American college freshmen in their first semester completed the study and were included in the analysis. The majority of the sample identified as female (*n* = 35). Of the 50 participants, 90% indicated they were 18 years of age (*n* = 45) and the remaining participants were 19 years of age. All of the participants indicated they were African American or Black and of the 50 participants, 3 participants indicated they were more than one race including White (*n* = 2) and Unknown (*n* = 1). Participants’ BMI values indicated that 2% of the sample was underweight (<18.5 kg/m^2^), 54% normal weight (18.5–24.9 kg/m^2^), and 28% overweight (25.0–29.9 kg/m^2^), with the remaining 16% classified as obese (≥30.0 kg/m^2^). All but one participant lived on campus (98%).

### 3.2. Data Availability

Of the 1656 EMA prompts that were delivered to participants, on 1421 (85%) occasions, the EMA prompt was answered. Three participants’ activPAL3 monitor malfunctioned resulting in complete data loss. Therefore, an additional 101 occasions were excluded from data analysis due to lack of accelerometer data. Finally, 10 occasions from 6 participants were not considered valid due to the activity monitor being worn for less than 60 min in the two hours prior to the EMA prompt. This resulted in 1310 occasions with valid EMA and activPAL3 data across 47 participants to be included in our final analytic sample. Participants, on average, provided approximately 28 occasions with valid EMA and accelerometer data (M = 27.9 occasions, Range = 9–34). An analysis of valid occasions indicated that EMA compliance (i.e., the likelihood of the participant missing an EMA prompt) did not differ by time of day, day of week, number of steps taken in the two hours prior to the EMA prompt, sex, or BMI (*p* > 0.05).

### 3.3. Descriptive Statistics

#### 3.3.1. Dietary Intake

On approximately 54% of EMA occasions (*n* = 707 occasions), participants indicated consuming food within the past two hours. On approximately 66.3% of EMA occasions (*n* = 869 occasions), participants indicated consuming a drink within the past two hours. Details on the prevalence of different types of food and fluid consumption are displayed in Table 1.

#### 3.3.2. Physical Activity

In the two hours before the EMA prompt, participants’ step count ranged from 0 to 8818 steps. On average, participants engaged in approximately 1300 steps in the two hours before the EMA prompt (M = 1279, SD = 1247). On 8.6% of occasions (*n* = 113 occasions) participants took no steps in the two hours prior to the EMA prompt. The intraclass correlation coefficient for step counts in the two hours before the EMA prompt was 0.07, indicating that the high degree of variability in step counts was due to within-person variability. This high degree of within-person variability indicates that it was appropriate to disaggregate step counts into between- and within-person components for multilevel modeling.

### 3.4. Multilevel Modeling

Results from multilevel models predicting the odds of food consumption are displayed in Table 2. There was no difference in the odds of consuming fruits (OR = 1.23, 95% CI: 0.39, 3.93, *p* = 0.71) or vegetables (OR = 1.32, 95% CI: 0.51, 3.40, *p* = 0.56) for participants who took more steps in the two hours before the EMA prompt, on average, compared to those who took fewer steps in the two hours before the EMA prompt, on average (i.e., null between-person associations). Significant within-person associations indicated that on occasions when participants took more steps than was typical for them in the two hours prior to the EMA prompt, they were more likely to report consuming fruits (OR = 1.43, 95% CI: 1.18, 1.73, *p* < 0.001) and vegetables over that same period (OR = 1.18, 95% CI: 1.02, 1.37, *p* < 0.001).

The odds of consuming fried fast food (OR = 0.64, 95% CI: 0.30, 1.34, *p* = 0.24) and chips/fries (OR = 1.75, 95% CI: 0.97, 3.15, *p* = 0.06) were not significantly related to the average number of steps taken in the two hours prior to the EMA prompt (i.e., null between-person associations). On occasions when participants took more steps than was typical for them in the two hours prior to the EMA prompt, they were more likely to report consuming fried fast food over that same period (i.e., significant within-person association; OR = 1.21, 95% CI: 1.01, 1.46, *p* = 0.04). The number of steps taken in the two hours prior to the EMA prompt on a given occasion was not related to the odds of consuming chips/fries (OR = 1.10, 95% CI: 0.95, 1.26, *p* = 0.17; i.e., null within-person association).

The odds of consuming sweets or pastries were positively associated with the number of steps taken at the between-person level but not the within-person level. Individuals who, on average, took more steps in the two hours prior to the EMA prompt tended to have higher odds of consuming sweets or pastries (OR = 2.76, 95% CI: 1.26, 6.025, *p* = 0.01). The number of steps taken on a given occasion was unrelated to the odds of consuming sweets or pastries (i.e., null within-person association; OR = 1.06, 95% CI: 0.93, 1.21, *p* = 0.32).

Results from multilevel models predicting the odds of fluid consumption are displayed in Table 3. There was no difference in the odds of consuming water (OR = 1.06, 95% CI: 0.41, 2.74, *p* = 0.89) or sugar-sweetened beverages (OR = 1.07, 95% CI: 0.55, 2.11 *p* = 0.84) for participants who, on average, tended to take more steps than those who tended to take fewer steps in the two hours before the EMA prompt (i.e., null between-person associations). On occasions when participants took more steps than was typical for them in the two hours prior to the EMA prompt, they were more likely to report consuming water (OR = 1.27, 95% CI: 1.13, 1.42, *p* < 0.001) and sugar-sweetened beverages (OR = 1.36, 95% CI: 1.22, 1.52, *p* < 0.001) over that same two-hour period (i.e., significant within-person associations). Sensitivity analysis of within-person associations between step counts and sugar-sweetened beverage consumption was driven by significant, positive within-person associations between step counts and fruit juice (OR = 1.29, 95% CI: 1.13, 1.46, *p* < 0.001), soda/sweet tea (OR = 1.24, 95% CI: 1.07, 1.44, *p* < 0.01), and sport drinks (OR = 1.32, 95% CI: 1.02, 1.70, *p* = 0.03) but not energy drinks (OR = 1.42, 95% CI: 0.77, 2.62, *p* = 0.25).

## 4. Discussion

This study is the first to use EMA and accelerometers to examine associations between device-measured physical activity and dietary intake behaviors in the context of everyday life in African Americans, a population disproportionately at risk for obesity and cardiometabolic conditions, during an important life transition, the first semester of college. Results from this study are important as they highlight the clustering of physical activity and dietary behaviors in real time. Results indicated that greater momentary physical activity (controlling for usual or average levels of physical activity over that same time) is associated with increased odds of consuming healthy dietary intake including water, fruits and vegetables, as well as increased odds of consuming unhealthy dietary intake including sugar-sweetened beverages and fried fast food. In other words, taking 1000 more steps than one normally would in a two-hour period increased the odds of consuming certain types of healthy food options over that same time period by 19% (vegetables) to 44% (fruits) but also increased the odds of consuming certain types of unhealthy food options over that same time period by 21% (fried fast food) to 36% (sugar-sweetened beverages).

By using intensive longitudinal data collection methods to capture physical activity and dietary behaviors in the real world, in real time, findings from this study provide insights into associations between physical activity and dietary behaviors. A recent longitudinal study examined changes in dietary intake as well as physical activity using summary-based measures over the transitions from high school to college and found that overall levels of typical physical activity (i.e., sport participation and active transport) decreased and typical consumption of fruits and vegetables, carbonated sugared soft drinks, sweets, and chips also decreased [36]. Though these results suggest that changes in diet and physical activity co-occur on macro-timescales (e.g., weeks, months, years), they did not provide insights into whether physical activity and dietary behaviors co-occur on micro-timescales (e.g., minutes, hours, days). As a result, it is unclear whether the between-person association among typical physical activity and dietary intake reflects temporally proximal behavioral coupling or is an artifact of aggregation over time. This study provides some of the first evidence that physical activity and dietary behaviors occur in temporal proximity to one another in college students (even after controlling for usual or average behavior over that same time) and rules out the possibility that previously reported associations [7] have been an artifact of aggregating across time and as a result linking temporally discontinuous behaviors (i.e., occasions of physical activity occurring at different moments or days than food or beverage consumption).

Recent research has documented the co-occurrence of within-person physical activity and dietary behaviors in samples of children and adults [16,17,37,38]. For instance, in an EMA study of overweight and obese adults, Crochiere et al. concluded that greater total physical activity (including device-measured light-, moderate-, and vigorous-intensity physical activity) in the time between EMA prompts was associated with a greater likelihood of reporting a dietary relapse (e.g., consuming energy dense food one intended to avoid) at an EMA prompt [17]. A study among 8–12-year-old children and their mothers (age 26–57 years) concluded that children were more likely than their mothers to consume unhealthy foods but equally likely to consume healthy foods during a two-hour period that also included physical activity [37]. Furthermore, findings by Dunton et al. concluded that on days when children engaged in more moderate- to vigorous-intensity physical activity than was typical for them, a greater percentage of their total calories on that same day were from carbohydrates (source of carbohydrates was not distinguished) but a smaller percentage of their total calories came from fat [16]. Additionally, children’s moderate- to vigorous-intensity physical activity on a given day was associated with a greater glycemic load on that same day [16]. These findings, in conjunction with findings from this study, suggest that physical activity co-occurs with other health-protective, but also health risk, dietary behaviors that have implications for weight maintenance or gain and ultimately obesity risk.

Findings that momentary physical activity is positively associated with unhealthy dietary intake (i.e., sugar-sweetened beverages, fried fast food) over that same time period may seem counterintuitive; however, there are several possible explanations for this. One possible explanation is that the act of planning and engaging in physical activity may deplete self-regulatory resources and reduce self-control, making it difficult to resist tempting food or drink options encountered immediately following physical activity [39,40]. Additionally, college campus food environments provide easy access to unhealthy food options in the cafeteria or other on-campus locations which may contribute to depleting self-regulatory resources and reduced self-control [41]. A second possible explanation is that physical activity and unhealthy food consumption have a compensatory relationship where college students consume unhealthy foods as a reward for engaging in physical activity. However, it is unclear whether these compensatory dietary behaviors add discretionary calories (i.e., individuals have calories to spare as they are below their Estimated Energy Requirements) or whether calories from these dietary intake behaviors lead to individuals exceeding their caloric needs. Alternatively, physical activity may be engaged in to offset calorie-rich, unhealthy foods such as sugar-sweetened beverages or fried fast food.

Yet, momentary physical activity was also positively associated with healthy dietary intake (i.e., fruits, vegetables, water), providing evidence that momentary health-protective behaviors can co-occur as well. Previous research has documented that engaging in physical activity has been linked to higher levels of dietary quality, including fruit and vegetable consumption, among college students on macro-timescales [10,42]; however, this is one of the first studies to document this co-occurrence on micro-timescales. It is hypothesized that many lifestyle behaviors are controlled by the same neural networks and thus engaging in one health-protective behavior can result in corollary engagement in another health-protective behavior [43].

In the present study, physical activity was operationalized as the number of steps taken in the two hours prior to the EMA prompt. There are many strengths of operationalizing physical activity in this way (e.g., steps are a basic unit of locomotion behavior, are easy to measure, and associated with clinically meaningful outcomes [29,30,31,32]). Despite this, future research focusing on the intensity of physical activity is also crucial to understanding momentary relations between physical activity and dietary intake. Our sample engaged in little moderate- to vigorous-intensity physical activity in the two hours prior to the EMA prompt (80% of observations had 0 min and 8% of occasions had more than 1 min of moderate- to vigorous-intensity physical activity). Future research examining these momentary associations may want to exclude participants self-reporting no moderate- to vigorous-intensity physical activity within the last week to enhance the likelihood of capturing engagement in moderate- to vigorous-intensity physical activity during time windows around the EMA prompt.

An important direction for future research is to understand how within-person associations between physical activity and dietary intake differ by weight status. This study was designed to detect within-person associations between physical activity and dietary intake. Therefore, the relatively small sample size prevented us from examining differences in within-person associations by weight status (i.e., a between-person moderator). Obese adults are more likely to consume energy dense foods such as fried foods as well as less likely to engage in physical activity compared to their normal weight peers [44,45]. Although these associations are documented at the between-person levels, it would be worthwhile to determine whether within-person associations between physical activity and dietary behaviors differ by weight status in future research.

Key findings from this study have important implications for intervention development. Interventions aiming to target individual health behaviors should be cognizant of potential complementary changes in health-protective behaviors while recognizing the potential for unintended, adverse changes in health risk behaviors. Additionally, understanding these within-person relations is key for designing effective and efficient multiple behavior change interventions. However, future research is necessary to determine the causal sequence of health behavior clustering as well as the extent to which this clustering impacts obesity risk and obesity-related chronic conditions.

This study has many strengths including the use of EMA and accelerometers to study the co-occurrence of momentary physical activity and dietary intake behaviors in the context of everyday life in an at-risk and understudied population. Despite these strengths, the limitations of this work should be addressed. Regarding measurement, the dietary EMA questionnaires did not assess how food was prepared (e.g., fried, baked, steamed) or the amount of food consumed. Though our measures were similar to previous studies employing a signal-contingent EMA protocol to assess dietary intake in various populations [28], and these assessments have been validated in college students as a measure of dietary intake [18], obtaining information regarding food preparation or total volume consumed can more accurately describe the nutritional value of what was consumed. Furthermore, this study did not assess other important information that could shed light on the context of physical activity or dietary behaviors (e.g., where university students ate or were active, who they were with). Accounting for contextual factors surrounding physical activity (e.g., intensity) and dietary intake (e.g., location) may also help to determine whether individuals took more steps to arrive at locations where food was available. Our within-person approach documented occasions when individuals took more steps than was typical for them (i.e., controlling for usual or average levels of physical activity), suggesting that associations in this study are likely not the result of an individual, for example, always walking to the cafeteria and drinking a sugar-sweetened beverage while in the cafeteria, because these behaviors represent an individual’s typical patterns of behavior (i.e., between-person associations) not an individual’s occasion-level deviation in their own behavior (i.e., within-person association). This study is an essential first step in understanding how physical activity and dietary behaviors co-occur in everyday life. Further research on the contextual factors surrounding these health behaviors could help clarify these associations.

Regarding recruitment, this study used a passive recruitment strategy, where eligible participants were sent an email and given contact information to respond to if interested in participating. As a result, those who participate in this study may be inherently more interested in physical activity and dietary behaviors compared to the general population [46]. Despite this possibility, our sample was similar to national figures regarding BMI classification among African American college students [47]. By having a representative sample in terms of BMI, it suggests that engagement in weight-related health behaviors such as physical activity and dietary intake are also likely to be representative of the broader African American college student population. Additionally, while this study focused on an at-risk group of individuals, this study is unable to make conclusions regarding racial differences in the co-occurrence of physical activity and dietary intake behaviors. Future research should explore differences between momentary associations between these behaviors by racial and ethnic groups.

Regarding study design, EMA prompting occurred during specific periods of time during the day (i.e., 9:30–10:30 AM, 12:30–1:30 PM, 3:30–4:30 PM, 6:30–7:30 PM, 9:30–10:30 PM). Despite the potential to capture dietary intake between 7:30 AM and 10:30 PM with this approach, it is possible that eating or snacking occasions were missed. For instance, if prompts occurred at 1:00 pm and 4:00 PM, eating occasions between 1:01 PM and 1:59 PM would not be captured with this protocol. Event-contingent prompting where, for example, participants self-initiate a questionnaire immediately following an eating occasions, in conjunction with a device-based measure of physical activity, may help to overcome this limitation in future research. Finally, because physical activity and dietary intake data were collected over the same two-hour period, causal inferences cannot be made. The decision to pair physical activity data with the two-hour time frame of the dietary EMA items as opposed to operationalizing physical activity as step counts in the time before the EMA item time frame was to avoid physical activity overlapping with the previous EMA prompt. While documenting the co-occurrence of these behaviors in naturalistic settings is valuable for behavior change interventions designed to promote weight loss or maintenance, a critical next step for research is to better understand the casual pathways linking these health behaviors.

## 5. Conclusions

Overall, this study provides insights regarding the co-occurrence of momentary physical activity and dietary intake behaviors in the context of everyday life in a vulnerable population. Employing EMA methods to assess different types of healthy and unhealthy dietary intake as well as device-based measures of physical activity revealed that on occasions when African American college students were more active than usual, they were also more likely to consume healthy items (e.g., water, fruits, vegetables) but also more likely to consume unhealthy items (e.g., sugar-sweetened beverages, fried fast food) over that same time period. Health promotion efforts to prevent or reduce obesity as well as chronic conditions associated with obesity among African American populations need to consider the co-occurrence of physical activity and dietary behaviors highlighted in this study.

## Figures and Tables

**Table 1 nutrients-12-01360-t001:** Prevalence of food and fluid consumption across ecological momentary assessment (EMA) prompts.

	Prevalence of Consumption across All Valid EMA Responses (*n* = 1310) *n* (%)	Prevalence When Food Consumption Was Indicated in EMA Response (*n* = 707)	Prevalence When Fluid Consumption Was Indicated in EMA Response (*n* = 869)
Food consumption			
Fruits	73 (5.6%)	10.3%	-
Vegetables	148 (11.3%)	20.9%	-
Chips or Fries	163 (12.4%)	23.1%	-
Sweets or Pastries	201 (15.3%)	28.4%	-
Fast Food	145 (11.1%)	20.5%	-
Fried Fast Food	81 (6.2%)	11.5%	-
Fluid Consumption			
Water	580 (44.3%)	-	66.7%
Sugar-Sweetened Beverages	376 (28.7%)	-	43.2%
Soda or Sweet Tea	150 (11.5%)	-	17.3%
Energy Drink	3 (0.2%)	-	0.3%
Sport Drink	30 (2.3%)	-	3.5%
Fruit Juice	213 (16.3%)	-	24.5%

Note. Consumption of coffee, milk, diet soda, and alcohol was reported in less than 2% of EMA prompts and therefore excluded from the table.

**Table 2 nutrients-12-01360-t002:** Results from multilevel models predicting the odds of food consumption.

	Fruits OR (95% CI)	Vegetables OR (95% CI)	Fried Fast Food OR (95% CI)	Chips/Fries OR (95% CI)	Sweets/Pastries OR (95% CI)
Intercept	0.01 ^ǂ^ (0.01, 0.03)	0.02 ^ǂ^ (0.01, 0.05)	0.01 ^ǂ^ (0.00, 0.03)	0.02 ^ǂ^ (0.01, 0.05)	0.06 ^ǂ^ (0.03, 0.13)
BP Step Counts	1.24 (0.39, 3.93)	1.32 (0.51, 3.40)	0.64 (0.30, 1.35)	1.75 (0.97, 3.15)	2.77 * (1.27, 6.03)
WP Step Counts	1.44 ^ǂ^ (1.19, 1.74)	1.19 * (1.02, 1.37)	1.21 * (1.01, 1.46)	1.10 (0.96, 1.26)	1.07 (0.94, 1.21)
Weekend Day	0.92 (0.52, 1.63)	0.70 (0.46, 1.08)	1.35 (0.83, 2.22)	0.78 (0.53, 1.16)	1.00 (0.70, 1.42)
Time of Day	1.31 (0.91, 1.88)	1.97 ^ǂ^ (1.49, 2.61)	2.21 ^ǂ^ (1.51, 3.22)	2.15 ^ǂ^ (1.64, 2.81)	1.51 ^ǂ^ (1.20, 1.91)
Female	3.49 * (1.14, 10.67)	2.09 (0.87, 5.02)	1.32 (0.65, 2.67)	2.33 ^§^ (1.31, 4.15)	1.23 (0.60, 2.55)
BMI	1.04 (0.98, 1.11)	1.02 (0.97, 1.08)	0.92 ^§^ (0.86, 0.98)	1.00 (0.96, 1.04)	0.94 * (0.89, 0.99)

Note. Table includes 1310 occasions from 47 participants. Step count was rescaled by 1000. OR = odds ratio; 95% CI = 95% confidence interval; BP = between person; WP = within person; BMI = body mass index. * = *p* < 0.05, ^§^ = *p* < 0.01, and ^ǂ^ = *p* < 0.001.

**Table 3 nutrients-12-01360-t003:** Results from multilevel models predicting the odds of fluid consumption.

			Sugar-Sweetened Beverages Sensitivity Analysis			
	Water OR (95% CI)	Sugar-Sweetened Beverages OR (95% CI)	Fruit Juice OR (95% CI)	Soda/Sweet Tea OR (95% CI)	Sport Drinks OR (95% CI)	Energy Drinks OR (95% CI)
Intercept	0.36 ^§^ (0.17, 0.76)	0.13 ^ǂ^ (0.08, 0.24)	0.08 ^ǂ^ (0.04, 0.16)	0.02 ^ǂ^ (0.01, 0.07)	0.00 ^ǂ^ (0.00, 0.01)	0.00 ^ǂ^ (0.00, 0.04)
BP Step Counts	1.06 (0.41, 2.74)	1.07 (0.55, 2.11)	0.88 (0.40, 1.95)	0.85 (0.27, 2.67)	6.84 * (1.50, 31.32)	6.70 (0.54, 82.90)
WP Step Counts	1.27 ^ǂ^ (1.13, 1.42)	1.36 ^ǂ^ (1.22, 1.52)	1.29 ^ǂ^ (1.14, 1.46)	1.24 ^§^ (1.07, 1.44)	1.32 * (1.02, 1.70)	1.42 (0.77, 2.62)
Weekend Day	0.60 ^ǂ^ (0.45, 0.80)	1.12 (0.84, 1.49)	0.92 (0.65, 1.31)	1.12 (0.75, 1.68)	2.20 * (1.01, 4.79)	1.32 (0.12, 15.03)
Time of Day	2.08 ^ǂ^ (1.73, 2.50)	1.66 ^ǂ^ (1.38, 2.00)	1.31 * (1.05, 1.64)	1.81 ^ǂ^ (1.38, 2.39)	1.71 (0.96, 3.05)	0.62 (0.14, 2.84)
Female	0.94 (0.41, 2.14)	1.41 (0.77, 2.58)	1.43 (0.71, 2.90)	1.35 (0.47, 3.87)	1.48 (0.37, 5.88)	0.89 (0.08, 9.98)
BMI	1.02 (0.97, 1.08)	0.96 * (0.92, 1.00)	0.97 (0.92, 1.01)	0.92 * (0.85, 0.99)	1.06 (0.98, 1.15)	1.02 (0.86, 1.22)

Note. Table includes 1310 occasions from 47 participants. Step count was rescaled by 1000. OR = odds ratio; 95% CI = 95% confidence interval; BP = between person; WP = within person; BMI = body mass index. * = *p* < 0.05, ^§^ = *p* < 0.01, and ^ǂ^ = *p* < 0.001.

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
