# Peer review of "Momentary Physical Activity Co-Occurs with Healthy and Unhealthy Dietary Intake in African American College Freshmen"

_nutrients, 2020, doi:10.3390/nu12051360_

Round 1

Reviewer 1 Report

This is a well written manuscript reporting on the results of a study which used ecological momentary assessment (EMA) and accelerometry to determine within-day, momentary associations between physical activity and food/beverage intake among African American college freshmen. The authors' expertise in EMA and statistical analyses is evident. The study does present a novel approach to measuring the co-occurrence of physical activity and dietary behaviors at the micro-timescale. Given the population and the environment in which the study was conducted, I do have some concerns ranging in level of importance. 

  • Lines 34 and 35: Use of the word "suffer" to describe the experience of people with obesity. Suffering is subjective and may not accurately reflect the experience of all people with obesity. Consider replacing with "are affected by". 
  • Line 84: Delete comma after "Despite,..."
  • Lines 173 - 192: Missing from manuscript.
  • Line 193: Subheading 2.3.2 missing.
  • Line 193 - 197: The activPAL allows for the intensity of an individual's steps to be quantified. Curious why the author's chose to use step counts alone when light-, moderate-, and vigorous-intensity would have provided more meaningful information?
  • Line 203: Considering the population, the author's should have collected information on living arrangements (e.g., on-campus vs off-campus). If this information is available, it should be reported and factored into the analyses as this may have implications for both momentary physical activity and dietary behaviors. 
  • Lines 223 - 234: It seems the sample size was too small for a logistic regression; otherwise, the study risks being under-powered. The authors should report the results of their power calculation and/or address the sample size in the limitations. 
  • Lines 339 - 341: I don't think the results of the current study provide a more nuanced understanding of the associations between physical activity and dietary behaviors. This study does not indicate where the students ate (home vs food court on campus), what kind of physical activity they were doing, or how intense the physical activity was. At most, we know students walk more around the time they report eating/drinking. 
  • Lines 373 - 395: The authors avoid mentioning the likeliest of explanations considering the population and environment for the study: The positive association between momentary physical activity and unhealthy and healthy dietary intake is because regardless of what participants ate or drank, they had to walk to and from a location (e.g., cafeteria or good court) to gain access to it. This would be easy to confirm if the authors know where the students regularly access food/drink and what the intensity-level of the steps are. 
  • Line 408: "...adverse changes [in] health risk behaviors."
  • Line 413 - 435: While the use of an accelerometer is a strength of the study, the decision to use steps counts alone when other aspects of physical activity could have been used (e.g., intensity) is a limitation that should be acknowledged.
  • Line 423: "additionally" should be additional 
  • Line 452 - 457: I disagree with the conclusion the results reveal a complex relationship between physical activity and dietary behaviors among African American students. At most, it confirms what we would expect of all students, regardless of race: walking occurs in the same time frame as food and beverage intake. Had the study included students of other racial/ethnic backgrounds, then we might have learned there are racial/ethnic differences in food/beverage intake in the same time frame physical activity occurs. That would be important to explore further. 

Reviewer 2 Report

This article explores the relationship between ‘in the moment’ physical activity and food and beverage consumption in African American college students. The authors report that higher step count in the previous 2 hours is related to higher consumption of both healthy and unhealthy foods. The study provides important preliminary data exploring the complexity of the relationship between PA and diet behavior. The authors address the common issue in dietary measurement of recall bias by reducing the recall period that is typically used (e.g. 7 days) down to only 2 hours using the ecological momentary assessment method. This is an innovative approach and the data will inform future research in this area.

Congratulation on a well conducted and interesting study! The article is very well written in terms of style and grammar and I commend you for your novel approach to dietary assessment and for focussing on the interrelationship with PA. I believe the article will be of interest to the readership of Nutrients. I just have a few minor suggestions which I believe will improve the manuscript:

General Comments:

  • In my copy of the manuscript lines 173-192 are missing
  • I was interested to know if mean total daily step count (a proxy of physical activity level) was associated with higher fast food/sugary beverage consumption overall. While I understand that you are looking at the immediate relationship between PA and diet (within a 2 hour window), there may be more transient effects occurring between PA and food consumption. Perhaps it is outside the scope of this paper, but it might be nice to include?
  • Could you explain how the accelerometer data were processed? Perhaps this is the section that is missing?
  • Were you able to control for any other demographic variables that might influence both physical activity and dietary habits?
  • In Table 1, not all beverages are listed. Perhaps there was a 0% response to alcohol/coffee etc? If so, it would be useful to clarify this in the table.
  • Just out of interest, why did you choose to measure physical activity with the activpal and not the movisens device which connects to the EMA software?

Round 2

Reviewer 1 Report

It appears you have addressed my concerns in the revised manuscript. You elaborated on the limitations of the study, which will be helpful to those interested in continuing this line of research. I think my concerns have been satisfactorily addressed. I’ll echo my earlier sentiment regarding how well the manuscript is written. I hope your findings spur further research in this area.